# Numerical Investigation of 3D Flow Properties around Finite Emergent Vegetation by Using the Two-Phase Volume of Fluid (VOF) Modeling Technique

**Amina [1] and Norio Tanaka [2],***

1    Graduate School of Science and Engineering, Saitama University, 255 Shimo-Okubo, Sakura Ku, Saitama Shi 338-8570, Saitama, Japan; amina168civil@gmail.com
2    International Institute for Resilient Society, Saitama University, 255 Shimo-Okubo, Sakura Ku, Saitama Shi 338-8570, Saitama, Japan
*    Correspondence: tanaka01@mail.saitama-u.ac.jp

**Abstract:** This study predicts how the Free Surface Level (FSL) variations around finite length vegetation affect flow structure by using a numerical simulation. The volume of fluid (VOF) technique with the Reynolds stress model (RSM) was used for the simulation. Multizone Hexahedral meshing was adopted to accurately track the free surface level with minimum numerical diffusion at the water–air interface. After the validation, finite length emergent vegetation patches were selected based on the aspect ratio (*AR* = vegetation width-length ratio) under constant subcritical flow conditions for an inland tsunami flow. The results showed that the generation of large vortices was predominated in wider vegetation patches (*AR* > 1) due to the increase and decrease in the FSL at the front and back of the vegetation compared to longer vegetation patches (*AR* ≤ 1), as this offered more resistance against the approaching flow. The wider vegetation patches (*AR* > 1) are favorable in terms of generating a large area of low velocity compared to the longer vegetation patch (*AR* < 1) directly downstream of the vegetation patch. On the other hand, it has a negative impact on the adjacent downstream gap region, where a 14.3–34.9% increase in velocity was observed. The longer vegetation patches (*AR* < 1) generate optimal conditions within the vegetation region due to great velocity reduction. Moreover, in all the *AR* vegetation cases, the water turbulent intensity was maximum in the vegetation region compared to the adjacent gap region and air turbulent intensity above the FSL, suggesting strong air entrainment over this region. The results of this study are important in constructing vegetation layouts based on the *AR* of the vegetation for tsunami mitigation.

**Keywords:** coastal vegetation; tsunami; aspect ratio (AR = vegetation width-length ratio); volume of fluid (VOF); Reynold's stress modeling (RSM); meshing

## 1. Introduction

Coastal vegetations play an essential role in disastrous floods such as tsunamis [1]. These vegetations are beneficial to minimize the velocity and energy of tsunamis [2]. The importance of coastal vegetation emerged after the 2004 Indian Ocean Tsunami (IOT) and 2011 Great East Tsunami Japan (GEJT), and various field investigations have been performed on the effectiveness of coastal vegetation in reducing tsunami flow energy [3,4]. To compare and measure the efficiency of coastal vegetation against field research, different experimental studies have also been carried out. The use of coastal vegetation as a tsunami countermeasure depends upon different factors, including density of vegetation [5,6], species [7,8], dimensions [9], alignments, and scale of vegetations [10]. Iimura and Tanaka [11] investigated the impact of tree vegetation density and found that the increment in vegetation density reduces the water level and velocity on the downstream side of the vegetation. Similarly, according to Pasha and Tanaka [12], the dense emergent infinite vegetation is more feasible for inland tsunami mitigation compared to the sparse emergent

infinite coastal vegetation, but due to the land constraints in inland regions [13], building infinite dense coastal vegetation in the actual field is not realistic [14]. To replicate the real scenario, a finite length coastal vegetation patch for inland areas should be provided [15,16]. Therefore, to safeguard inland regions from catastrophic tsunami flow, a thorough investigation must be conducted to examine the flow changes surrounding finite length coastal vegetation. Although previous researchers have investigated the utilization of finite coastal vegetation with different gaps [17] and varying ground slopes through two-dimensional (2D) numerical simulation [18], some authors have studied in detail the behavior of flow analysis around discontinuous vegetation patches [19,20], while the authors in [21–25] studied internal flow characteristics and turbulent parameters without taking free surface movement into account. However, there has been less numerical research on the impact of three-dimensional (3D) flow behavior with free surface movement around a limited coastal vegetation patch by changing its aspect ratio (dimensions), i.e., length and width, for tsunami prevention. The results of this research can be used to better understand the role of inland coastal vegetation of a finite length in the event of major floods such as a tsunami by allowing researchers to compare flow characteristics between longer and wider vegetation patches. Hence, the present study is mainly focused on the numerical investigation of 3D flow behavior through longer (AR < 1) and wider (AR > 1) vegetation patches to determine the flow behavior of each kind with their flow affecting properties, based on the Aspect Ratio (AR = vegetation patch width/length) of the finite length vegetation. Furthermore, the research also includes a vast scope and broad applicability in clarifying the flow mechanism in the form of complex velocity patterns within and around the finite length vegetation.

The velocity patterns are nearly impossible to capture through an experimental investigation due to the limitations of instruments: laser sheets in a Particle Image Velocity Meter (PIV), for example, are not viable in the vegetation zone due to the presence of vegetation cylinders. Additionally, velocity sensors in the Electromagnetic Flow Meter (EFM) cannot be utilized as they can disturb the flow near the vegetation region and the bed region. This numerical study provides an advantage in bridging this gap. To accomplish the objectives through numerical investigation, we used a CFD tool called FLUENT, which runs a simulation using the Volume of Fluid model (VOF) with the Reynolds stress model (RSM). The flow properties are addressed in terms of three-dimensional free surface variations, velocity distribution, velocity contours, velocity vectors, and turbulence characteristics.

## 2. The Volume of Fluid Model (Air–Water) Two-Phase Flow

In 3D numerical simulations, a multi-phase model such as the volume of fluid (VOF) model is typically used to treat the free surface boundary and interaction between the two-phase flow (water and air). The VOF technique was initially suggested by Hirt and Nickhols [26]. VOF modeling was used by the majority of earlier studies to solve flow issues in stepped spillways and wide crested weirs [27–31]. However, few studies have been performed using VOF modeling and provide solutions mainly to basic rectangular and curved open channel flow problems [32,33]. Therefore, this study considers the vegetated open channel to check the performance of the VOF modeling.

By solving a single momentum equation, the VOF approach can model two or more immiscible fluids and control the fraction of each of the volume of the fluid in the entire domain. The fundamental principle of this kind of model is that two fluids do not interpenetrate. In this problem, the use of two fluids creates a new variable, i.e., the computational cell phase volume fraction. Consequently, regarding the field of air and water flow, a single set of air and water share a momentum equation and in each computational cell, the volume fraction of each fluid is followed across the whole domain. The volume fraction for all phases in each computational cell add up to (1) unity. The flow structure and flow properties for all variables are separated into phases and are indicated as volume-averaged values, such that the volume fraction of each phase is known at each location. Therefore, the variables and properties in each cell, depending upon the volume fraction value, are

purely representative of a mixture of the phases. If the volume fraction of the $q_{th}$ fluid in the cell is referred to as $a_q$, then the following three conditions are possible:

(a)  $a_q = 1$ when the $q$th fluid fills the whole cell demonstrating that the cell is filled with water.
(b)  $a_q = 0$ when the $q$th fluid does not occupy the whole cell means that the cell is filled with air.
(c)  $0 < a_q < 1$ indicates that where the cell has an interface between the $q$th fluid, i.e., free surface between air and water layer.

If $a_a$ = air volume fraction and $a_w$ = water volume fraction, the free surface traceability of the water–air interface is performed by a continuity equation solution is provided by:

$$\frac{\partial \alpha_w}{\alpha_t} + \frac{\partial \alpha_w}{\partial x_t} = 0 \tag{1}$$

To evaluate the precise orientation of the free surface, i.e., the water–air interface, the VOF implicit scheme is used, and within each cell it has a linear slope, which is used to measure fluid advection via the corresponding cell face. The location of the linear interface to the center of each partially filled cell is therefore measured by the value of the volume fraction and its derivative within the cell.

## 3. Turbulence Modeling (RANS)

The Reynolds-averaged Navier–Stokes (RANS) equations, written in tensor notation, are used to define the flow behavior in vegetated channels.

$$\frac{\partial \overline{u}_i}{\partial x_i} = 0 \tag{2}$$

$$\frac{\partial \overline{u}_i}{\partial t} + \frac{\partial}{\partial x_j}\left(\overline{u_j u_i}\right) = -\frac{\partial \overline{p}}{\partial x_i} + v\frac{\partial^2 \overline{u}_i}{\partial x^2_j} - \frac{\partial}{\partial x_j}\left(\overline{u'_i u'_j}\right) \tag{3}$$

where overbar represents the Reynolds averaged value and $\overline{u}_i$ and $\overline{u}_j$ are the average and fluctuation velocities in the $x_i$ direction, respectively. I = (1, 2, 3), indicating the longitudinal $x$, transverse $y$, and vertical $z$ axis. Equations (2) and (3) may be calculated for the average value of velocity, pressure, and other variables when the turbulent correlations can be linked to average velocity or mean pressure, respectively. This is referred to as the turbulence modeling closure problem. Eddy viscosity, a concept developed by Boussinesq's, was utilized to simulate Reynold's stress, described by.

$$-\overline{u'_i u'} = v_e\left(\frac{\partial \overline{u}_i}{\partial x_j} + \frac{\partial \overline{u}_j}{\partial x_i}\right) - \frac{2}{3}k\delta_{ij} \tag{4}$$

In which, $v_e$ represents the eddy/turbulent viscosity and $K = 0.5\overline{u'_i u'_j}$ denotes the turbulent kinetic energy (TKE). The eddy viscosity is unknown, but through the dimensionless analysis it is given by.

$$v_e = C_\mu\left(\frac{K^2}{\in}\right) \tag{5}$$

Here, $C_\mu = 0.09$ is the coefficient of closure and $\in$ represents the dissipation rate of turbulent kinetic energy (TKE).

## 4. Material and Methods

*4.1. Conditions for Numerical Simulation*

4.1.1. Flow Conditions

The tsunami, which is a series of waves, continues to the shore, and runs up inland with a huge discharge [34]. On the other hand, if discontinuous vegetation patches are

present in the path of flow, then the hydraulic conditions may change [35,36]. However, when the tsunami inundation enters in the inland regions, it does not contain any waves and propagates into the inland region with a long period. As a result, the tsunami inundation can be represented as a quasi-steady flow [37]. Many earlier researchers considered the flow in the inland area surrounding an inland vegetative forest in a steady and subcritical condition when modeling the tsunami flow. Thus, the flow parameters in this research were defined using Froude and initial water depth similarity, which assumes a subcritical and constant tsunami inland flow. During the 2004 Indian Ocean tsunami, the Froude conditions ($F_r$) were reported to be 0.64–1.04 in the inland region of Banda Aceh [38]. Furthermore, the tsunami flow around an inland forest in Miyagi Prefecture, Japan, was subcritical at numerous locations during the 2011 Great East Japan tsunami (GEJT), with varied Froude values in the range of 0.7–1 and an approximate tsunami depth of 7.3–8.3 m [39]. The vegetation is set to be placed inland in the present research. To simulate a tsunami flow on an actual scale, a model scale of 1/100 was used, and the Froude number in a channel was kept around 0.7 without any vegetation inserted in the rectangular channel. In this numerical research, the water depth (without a vegetation model) used to create the subcritical inland tsunami condition was 4.5 cm, against the initial Froude number of 0.70. Since the numerical domain used a two-phase model, i.e., (VOF), it is necessary to specify how much air is included above the water region. Salaheldin et al. [40] stated that there is no impact from the boundary at the top if air height to water height is kept above 0.5. In our current 3D simulation, the total air portion depth is maintained at 5.5 cm, resulting in a ratio of 1.22, which is sufficient to prevent any impact from the top of the domain boundary state.

### 4.1.2. Vegetation Conditions

The Japanese pine tree, with an average tree height of 15 m and a trunk diameter of 0.4 m, located in the Sendai Plain, was adopted as the tree species for the vegetation model. According to Tanaka et al., [41], a tree can be modelled as a rigid circular cylinder if the crown part of the tree is high above the tsunami height. The trees were thus modelled in 1/100 scale, utilizing rigid cylinders with a diameter of 0.4 cm placed in a staggered pattern, in accordance with the average diameter of the pine tree. The density of the vegetation is defined by S/d number [2]. There are three different vegetation densities based on this S/d number (where S represents the spacing between the rigid cylinders in a cross streamwise direction and d is the cylinder diameter). The sparse model is defined as having a S/d number of 2.13, while the intermediate and dense models have a S/d value of 1.03 and 0.25, respectively. In the current research, an intermediate density with a S/d value of 1.03 was used for the vegetation model. As stated in the introductory section, the construction of infinite vegetation in the inland areas would be impossible due to land constraints; therefore, to replicate the actual situation, a limited length vegetation was examined in this research to represent the inland vegetation. The main goal of the research was to examine the effect of change in the length and width of the vegetation model on the 3D flow characteristics, while accounting for free surface fluctuation based on the aspect ratio ($AR = W_y / W_x$, in which $w_x$ and $w_y$ are the vegetation length in the *x*-direction and width in the *y*-direction, respectively), which was considered as a changing parameter. Five different cases were considered with varying length and width of vegetation model, as shown in Figure 1. The vegetation models with five various configurations were placed separately in the channel to evaluate their performance in terms of flow resistance. The AR of the finite length vegetation in this study was in the range (0.2–5), as shown in Table 1. The AR of 0.2 and 0.3 indicate longer vegetation patches and the AR of 3 and 5 indicate wider vegetation patches configurations in the path of flow.

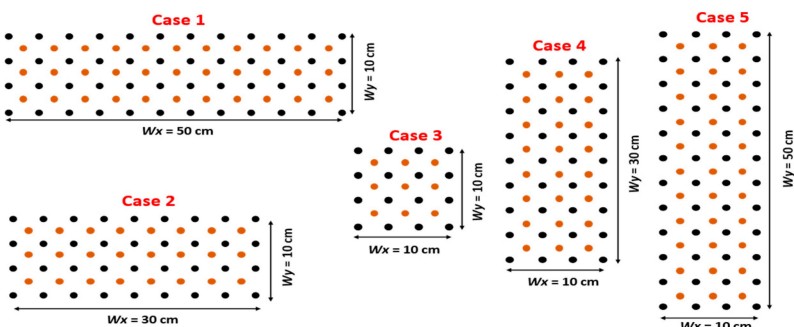

**Figure 1.** Layout of the domain for numerical simulation: Arrangement of the Vegetation patches.

**Table 1.** Hydraulic Conditions of Numerical Model: Note: $W_x$ represents vegetation length; $W_y$ depicts vegetation width; *Fr* shows Froude number ($Fr = U / \sqrt{gh}$, where $U$ represents the initial flow velocity, $g$ is the gravitational constant and h is the initial water depth); $Dw$ stands for water depth; $U$ is the initial velocity.

|  | Cases | AR | $W_x$ (cm) | $W_y$ (cm) | Fr | S/d | $Dw$ (cm) | U (m/s) |
|---|---|---|---|---|---|---|---|---|
| Longer Vegetation Patch | 1 | 0.2 | 50 | 10 | 0.7 | 1.03 | 4.5 | 0.465 |
|  | 2 | 0.3 | 30 | 10 | 0.7 | 1.03 | 4.5 | 0.465 |
|  | 3 | 1 | 10 | 10 | 0.7 | 1.03 | 4.5 | 0.465 |
| Wider Vegetation Patch | 4 | 3 | 10 | 30 | 0.7 | 1.03 | 4.5 | 0.465 |
|  | 5 | 5 | 10 | 50 | 0.7 | 1.03 | 4.5 | 0.465 |

4.1.3. Measurement Locations

A rectangular channel 150 cm long and 70 cm wide was used to represent the current numerical domain, which included a vegetation model that covered the finite width of the domain. The top view of the channel domain is presented in Figure 2. Six specified points were selected to investigate the vertical distribution of stream wise velocities within the vicinity of the vegetation model (*x*1, *x*2, *x*3) and the adjacent gap region (*y*1, *y*2, *y*3). Furthermore, an important horizontal surface (at depth of *z* = 3.5 cm below the initial water depth) and two longitudinal sections (LS1 = 35 cm passed through the centre of channel domain or vegetation model and LS2 = 25 cm, passed through the adjacent gap region throughout the channel domain),as well as two cross streamwise sections (CS1, located within the vegetation model and CS2, located downstream of the vegetation model) were considered for the detailed investigation of the flow properties in the form of free surface level (FSL) distribution, velocity profiles, velocity vectors (flow movement), velocity and turbulent intensity contour plot distribution.

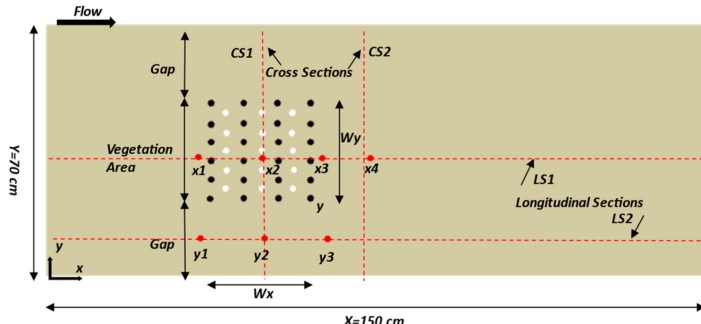

**Figure 2.** Top view of the vegetated channel setup and specified positions and sections. The red points depict the concerned locations, whereas the red dashed lines show the measurement sections.

*4.2. Preprocessing and Postprocessing*

4.2.1. Preprocessing

The simulation was carried out using the Computational Fluid Dynamics (CFD) tool, ANSYS. The numerical setup for this study consisted of two steps: preprocessing and postprocessing. In preprocessing, the geometry was modeled using a design modeler tool. A rectangular channel 150 cm long and 70 cm broad was created in a design modeler, together with a vegetation model (staggered arrangement) that occupied the domain's limited width. The arrangement of the vegetation model and the spacing in between the cylinders is shown in Figure 3a. The domain was then transferred to the mesh tool. Meshing is a critical stage in achieving high-quality results, as desired outcomes require proper meshing, which is described as the partition of the geometry into components, including cells and elements. Simulation precision, convergence and speed are all influenced by mesh size and density. More memory and time are required for fine meshing, whereas numerical diffusion in the results occurs when meshing is carried out incorrectly. Firstly, we tested various meshing methods (e.g., sweep meshing, tetrahedral primitive with numerous deformable components resulted in higher skewness values and in some cases negative volumes) on an open channel without vegetation to check the diffusion rate of water and air interface; however, we achieved convergence with reasonable computation time using multizone meshing with hexahedral components. Figure 3b,c depict the differences in results in terms of diffusion rate between the water and air volume fractions on a channel domain, when multizone and tetrahedral meshing are used separately. Tetrahedral meshing generated incorrect results and maximized numerical diffusion at the water–air interface, resulting in significant disparities. This was because tetrahedral meshing consisted of only tetra cells and did not well distribute the cells to define the boundary between air and water phase. It also created the poor-quality surface mesh around the vegetation cylinders. On the other hand, multizone meshing, which is a hybrid of hexahedral or brick and tetra elements, generally result in more accurate results at lower element counts than only tetra elements. Therefore, multizone meshing produces the best results with low diffusivity and can precisely track the water–air interface. Furthermore, the skewness and orthogonal quality criterion methods were used to evaluate the quality of the mesh. Skewness defines how close to the ideal (equiangular quad) a face or cell is. According to the definition, a value that corresponds to 0 indicates "Excellent", and 1 corresponds to the worst cells (degenerate) quality. For this present study, the equilateral volume-based skewness method was used to check the skewness. Under this method, the mesh is considered a good quality for 3D if its skewness value is less than 0.4. The present geometry mesh has a value of 0.16, which leads to excellent mesh quality. The mesh of the whole domain, especially around the cylinders, is shown in Figure 3d.

Finally, in order to acquire the best simulation results, a grid independency test was also conducted. The experimental results of vertical velocity distribution at position 1 (see Figure 3e were compared with the results of varying the mesh size of the numerical domain. Initially, the mesh grids of 0.4 million (coarse), 1.7 million (fine), and 2.8 million (finer) were investigated. A 7% change in vertical streamwise velocities between the coarse and fine grids was observed at position 1 and 2; however, the variance in findings by greater refinement was just 1%. Therefore, the fine mesh with 1.7 million grids was selected for the present study. It has been shown that the vertical stream wise velocity values for coarse grid are quite greater than those for medium and fine grid, in comparison with the experimental values, Figure 3e. This might be because a coarse mesh cannot properly recognize the boundaries of the vegetation cylinders, and so the resistance or drag produced by these structures could not be accurately calculated, resulting in significantly greater flow velocity magnitudes. On the other hand, a fine mesh (refinement method, particularly near the cylinders) solves this issue by estimating a realistic flow shape and ensuring computational precision.

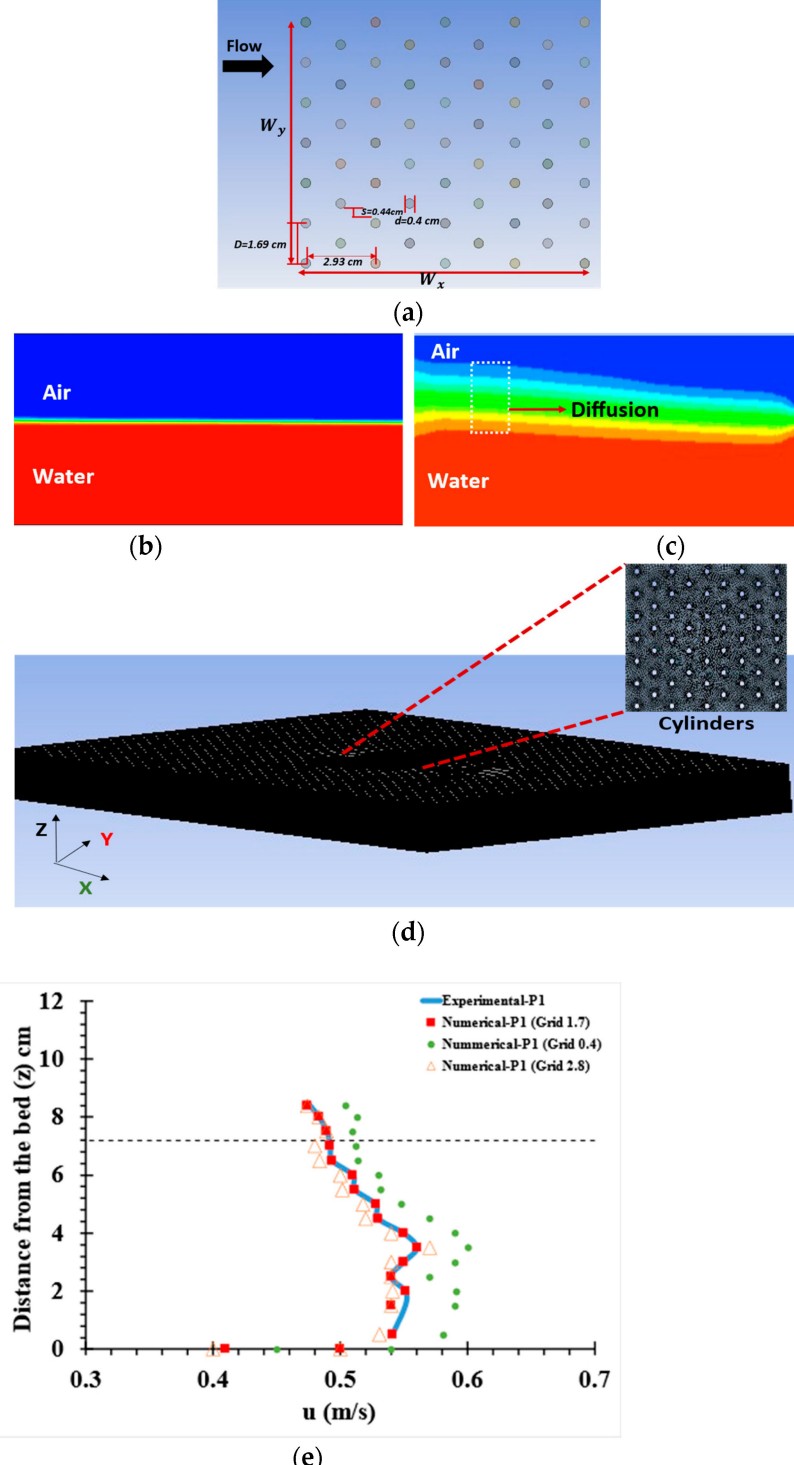

**Figure 3.** Preprocessing Setups. (**a**) Arrangement and spacing of cylinders: Comparison of Meshing techniques: (**b**) Multizone meshing (**c**) Tetrahedral meshing. (**d**) Mesh domain (**e**) Grid independency test.

### 4.2.2. Postprocessing

After meshing was complete, the next step was to set up the physics, which was carried out in post processing. In multiphase models, the volume of fluid (VOF) model was adopted, while the Reynolds stress model (RSM) was used for viscous modeling. The next step was to designate the phases; water liquid was selected as the primary phase, while air was defined as the secondary phase. The boundary conditions were than assigned to

the faces of the geometry The velocity inlet boundary condition was given to the water inlet, and the pressure inlet was applied to the air inlet. The water and air density were set at 1000 (kg/m³) and 1.225 (kg/m³), respectively. The top boundary and outlet were both considered as pressure outlets, with the gauge pressure set to zero. However, just one outlet was defined so that the solution could use a flow level derived from the inner flow field. The boundary condition applied to the bed of domain, faces of cylinders are considered as a no-slip wall. The no-slip rule for viscous fluids in fluid dynamics means that the fluid will have zero velocity at the solid boundary, as many researchers have used this condition [42–44]. For side walls, a slip boundary condition was applied to avoid the effects of the side wall. The boundary conditions applied to the whole domain is shown in Figure 4.

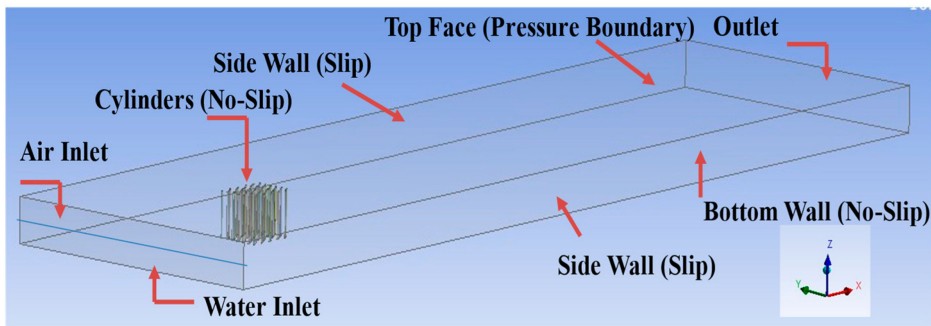

**Figure 4.** Postprocessing Setups: Boundary Conditions.

The solver for time was used as transient. The type for interface modeling for VOF was used as sharp/dispersed to avoid diffusion rate. Finally, for the pressure velocity-coupling method, a SIMPLE scheme was adopted and for spatial discretization, the third order upwind scheme was used. Under-relaxation factor values were considered as low for convergence criteria and residual smoothing. A fully implicit scheme was incorporated for the time incorporation. The normalized residual value was set at $1 \times 10^{-6}$; thus, when the solution's calculations were less than the specified number, it was considered converged. The standard initialization method was used for solution initialization. The user guide [45] contains descriptions of the algorithm, governing equations, and turbulence model.

## 5. Results and Discussion

### 5.1. Model Validation

A laboratory experiment was conducted to validate the results of the numerical model. The experimental setup was established in a laboratory channel of a rectangular cross-section with a length (*L*) of 500 cm, width (*w*) 70 cm, and a height of 50 cm at Saitama University, Japan. The vegetation was configured as an array of wooden cylinders in a staggered arrangement and precisely embedded on the channel bed at the center of the streamwise width, around 55 cm upstream of the channel. The dimensions of the vegetation model were chosen to be the same as those used in the current numerical model of Case No. 3 (Table 1), whereas the density of the vegetation model was considered sparse (*S/d* = 2.13). The boundary effect, also known as the side wall effect, has an influence on the channel flow structure when the blockage ratio, which measures how wide the vegetation patch front is in relation to how wide the channel itself is, exceeds a certain threshold value. According to previous studies, an utterly trivial impact of the side walls on the drag of a flat plate occurs when the obstruction ratio is equal to 5–6 percent [46]. To determine how the drag force decreases with increasing obstruction ratio (up to 40 percent), Okajima et al. [47] performed an experiment around a cylinder shaped like a rectangle, in which the flow rate was constant. They reported that the drag forces initially drop (obstruction ratio of 9 to 10 percent) and then rise as flow obstruction increases. The vegetation model (VM) studied in the present research has the capacity to enable water to flow through it, and this is mostly

based on the density. As a result, in order to compute the blockage ratio, the front width ($Wc$) of the vegetation was estimated by multiplying the number of cylinders in each of the first two rows due to its staggered arrangement by the cylinder diameter. Then, the front width of the vegetation patch is divided by the channel's width. The considered vegetation model for experimental trial had a blockage ratio up to approximately 13%, which was under critical limit. Furthermore, it was also observed during the testing that no waves were reflected from the channel's side wall into the area under investigation during the experimental trials. As a result, it was assumed that the boundary effect had no influence on the results and the effect of the restriction was overlooked.

The initial water depth was 7.2 cm, which corresponded to an initial Froude value ($Fr_o$) of 0.73. The free surface level (FSL) was measured by point gauge at an interval of every 20 cm longitudinally along the centerline from the channel upstream to just behind the vegetation. Two positions were chosen (P1, located at front of the vegetation model and P2, located at back of the vegetation model) to measure the vertical distribution of streamwise velocity using an Electro Magnetic Flow Meter (EMFM). The velocity measurements were obtained by positioning the EMFM slightly above the flume bed and raising it vertically at an interval of 0.5 to 1 cm up to the FSL. The experimental setup of the vegetation model (Measurement Locations) and the arrangement of cylinders and resulting flow structure are presented in Figure 5a,b. To mimic the experimental setup for numerical modeling and avoid a large mesh grid pattern and lowering computing costs, a vegetation model in a channel with a length of only 90 cm and a width of 70 cm was built. The preprocessing and postprocessing procedures used for the validation domain were the same as those used for the present numerical model.

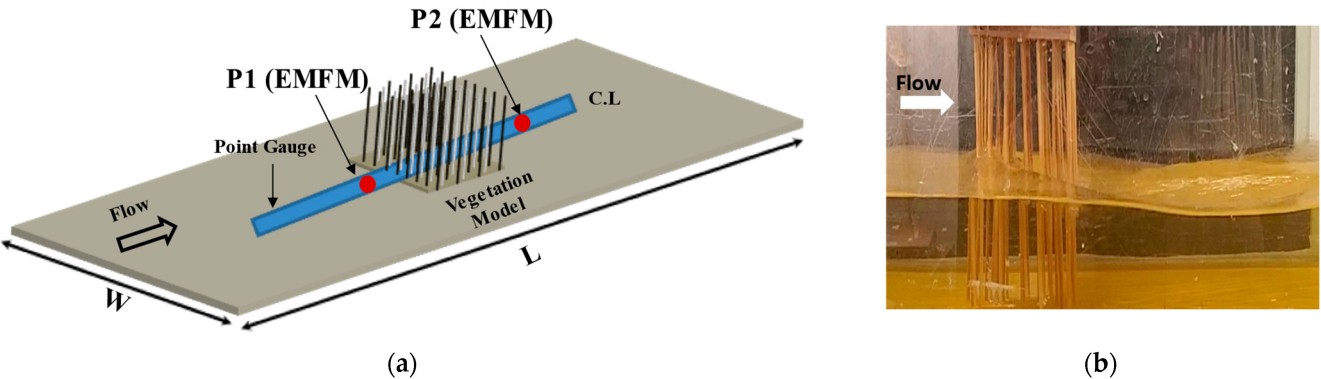

**Figure 5.** Model Validation (**a**) Experimental Flume setup and specific locations for measurement of water level and vertical distribution of streamwise velocity; (**b**) Resulting flow structure.

The longitudinal distribution of a FSL calculated over the center portion of the domain for both the experimental and numerical results are presented in Figure 6a. The ordinate indicates the FSL in cm, while the abscissa represents the longitudinal distance along the channel. After installing the vegetation cylinders in the channel, for both the experimental and numerical results, the FSL was raised on the upstream side due to the vegetation model retardation and followed a decreasing trend in the downstream region of the vegetation side. This difference in elevation of the FSL between upstream and downstream of the vegetation developed the slope of the FSL inside the vegetation. Furthermore, Figure 6b depicts the results of the vertical distribution of streamwise velocity at defined locations from both the experimental and numerical findings. The streamwise velocity was considerably lower at the back (P1) of the vegetation patch compared to the front (P2) of the vegetation patch in both experimental and computational outcomes. The low velocity magnitudes of velocity at the back of the vegetation were caused by the retardation or blockage by the vegetation cylinders, which is beneficial for the mitigation of high tsunami magnitude. The numerical model outcomes are consistent with the findings in the experiment, demonstrating the model's validity. However, near the bed region, a difference in velocities was observed

between the experimental and numerical findings. This discrepancy may be attributable to the fact that the EMFM's velocity sensor cannot be used in close proximity to the bed.

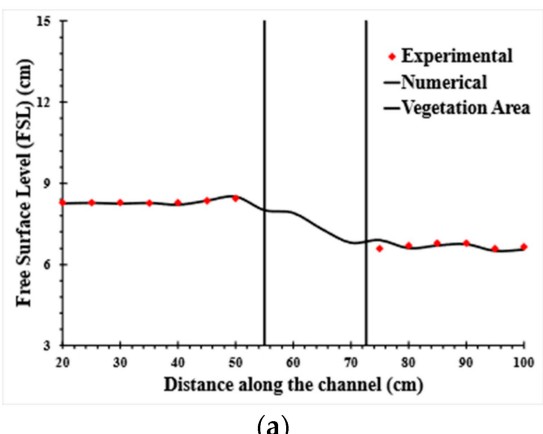

(**a**)

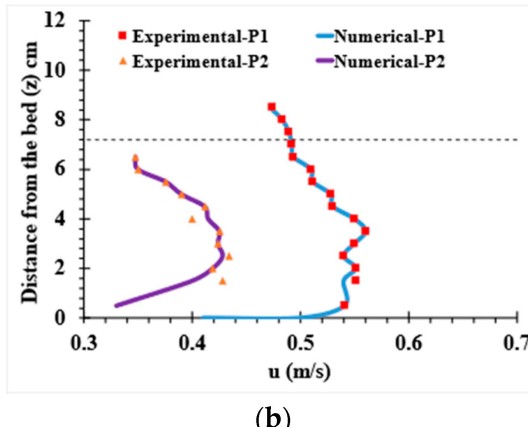

(**b**)

**Figure 6.** (**a**) Comparison of computed and experimental free surface level (FSL) profiles: (**b**) Comparison of computed and experimental vertical distribution of streamwise velocity.

*5.2. Flow Characteristics*

5.2.1. Free Surface Profiles

Vegetation in coastal areas is essential in reducing the velocity and energy of tsunami-generated floodwater. The flow properties around the coastal vegetation are highly dependent on external flow structure, i.e., free surface variations. Water reflection and vegetation resistance can reduce the flowing water energy, depth of inundation, region of inundation, and velocity behind the vegetation [12]. Therefore, this numerical study mainly focused on capturing the three-dimensional external flow structures, i.e., FSL variations around finite length emergent coastal vegetation of different configurations based on the AR and its impact on the flow behavior for tsunami mitigation. The two dimensional as well as three-dimensional vortices generation and free surface movement around the vegetation models of four cases (2, 3, 4, 5) are presented in Figure 7a–h. The free surface level (FSL) differences at the front, mid and behind the vegetation model are presented in 2D water–air volume fraction contours considering the longitudinal section (LS1), Figure 7a–d. The blue and red color represent the air and water, respectively. The 3D water volume fraction contours, i.e., Figure 7e–h are presented to understand the free surface vortices, overall shape, and the free surface wavy pattern in the vegetated open channel. The red color in the rectangular channel shows the water phase. By visualizing the contours, the formation of small and large-sized free surface vortices and their development process mainly depend on the vegetation elements configuration. The array of vegetation elements at the upstream side caused the flow to be irrotational. Retardation occurs due to the patch blockage impact and the drag interface discontinuity offered by the cylinders. The water gradually bleeds with minimum velocity through the vegetation area and travels rapidly at high velocity in the adjacent vegetation gap regions. The shear layer was formed by the significant difference in velocity between the vegetation patch and its gap area due to the FSL elevation difference. The shear layer then produces coherent structures or vortices due to Kelvin–Helmholtz instability. The vortices generated by the shear layer dominate the flux exchange (low and high velocity) between the vegetated and the gap region. The production of vortices and the exchange of mass flux at the interface between the vegetated and gap regions can be assumed to play a significant role in sediment transport into the vegetated plain [48]. These vortices have continuously grown from the upstream side towards the downstream side and gradually dissipate in all cases (2–5). In the longer vegetation patch (AR < 1), i.e., case (1) and case (2), it was found that the increment in elevation of FSL in front of vegetation was less and less steep slope was generated within the vegetation Figure 7a,b, resulting in the creation of vortices of small size vortices Figure 7e. Therefore, a longer

vegetation patch would provide a smaller contribution to the overall water flow energy dissipation capabilities. Similarly, in case (3), i.e., (AR = 1), small sized vortices were also found when the length and width were same, as in Figure 7c,f. In addition, wider vegetation patches (AR > 1), i.e., case (4) Figure 7g and case (5) Figure 7d,h produced considerable resistance due to increased vegetation cylinders in the cross streamwise direction, and hence, maximum elevation of the FSL was observed. According to the findings of the Pasha and Tanaka [16] study, increasing the width of the vegetation patch results in a greater depth of water toward the front side of the vegetation, demonstrating that the results of the current study are in agreement. The maximum FSL results in a steep gradient slope inside the vegetation patch and minimum elevation of FSL was found on the vegetation patch downstream side. This higher difference in FSL upstream and downstream side of the vegetation patch resulted in the development of strong large vortices, Figure 7g,h. The maximum FSL was observed in the maximum AR, i.e., case (5), around 66% compared to the initial FSL, while this increment was observed to be 53% in case (4). Contrarily, fewer percentage differences were observed in lower AR cases, i.e., cases 1, 2, and 3, around 25%, 22%, and 20%, respectively. Thus, in this study, the wider vegetation patches significantly increased the FSL at the front side of the vegetation patch, which would help dissipate the flow energy behind the vegetation patch. These dramatically changed FSL heights in the upstream, inside, and behind the vegetation, and the development of vortices at the free surface were difficult to observe in those numerical studies where they assume water as a flat surface [21–23]. Therefore, VOF modeling (water–air interface) can well predict the FSL around the vegetation in an open channel.

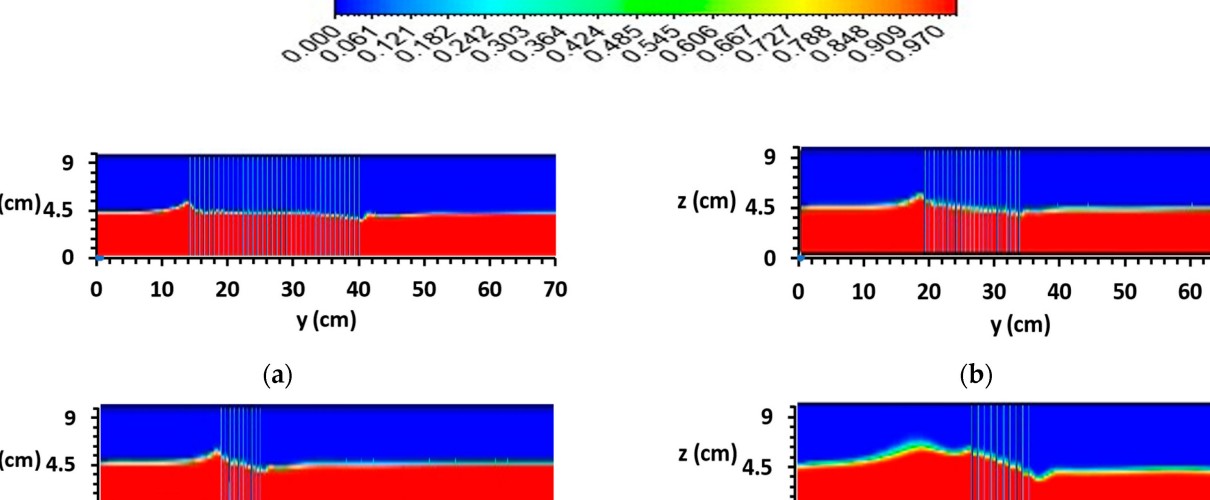

**Figure 7.** *Cont.*

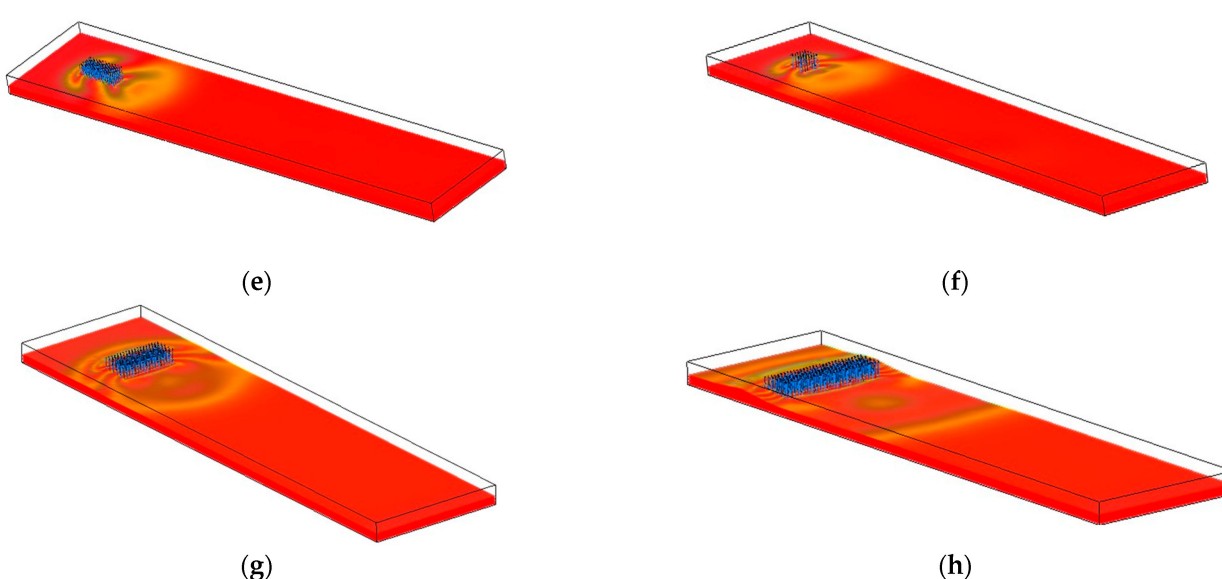

**Figure 7.** Two-dimensional (2D) Distribution of water–air volume fraction (**a**) Case 1, (**b**) Case 2, (**c**) Case 3, (**d**) Case 5: Three-dimensional (3D) distribution of water volume fraction (**e**) Case 2, (**f**) Case 3, (**g**) Case 4, (**h**) Case 5.

5.2.2. Velocity Distribution of Streamwise Velocity Profiles

The streamwise velocity distribution in the vertical direction around the finite length vegetation and their adjacent gap regions was investigated. The behavior of the streamwise velocity distribution is divided into three distinct regions that cover the whole array of vegetation cylinders in the vertical direction. (i) The region near the free surface is influenced by free surface oscillations. (ii) The middle region, found between the free surface and the bed region, where the vertical cylinders control the flow, and in that region, flow characteristics remain nearly constant in the vertical direction. (iii) The approaching bed region where the flow is extremely three-dimensional because of the bed interaction. The streamwise velocity at the centerline of the longitudinal section (L1) at the four specified points (X1–X4) for all the AR cases (1–5) is shown in Figure 8a–d. Point (X1) is located on the upstream side of the vegetation, and point (X2) is located at the center of the vegetation region. In addition, Point (X3) and (X4) are located at the vegetation region's downstream side. The x-axis represents streamwise velocity ($U_{veg}$), while the y-axis indicates distance from the bed ($Z_w$) to the FSL. Due to the presence of vertical stems, as well as the resistance given by the bed of the domain, the velocity is lower closer to the bed area of the domain. The maximum reduction in velocity was observed in wider vegetation patches (AR > 1), i.e., case (4) and case (5), at point (X1) due to the greater resistance provided by the cylinders in the crosswise direction and the increased height of the FSL, while less reduction in velocity was observed in the longer vegetation patch (AR < 1), i.e., case (1) and case (2). The results clearly demonstrate that wider vegetation patches (AR > 1) could effectively reduce the velocity in front of vegetation. In addition, near the free surface region in Figure 8a, the highest reduction in velocity was observed in comparison to the middle region, resulting from the difference in pressure caused by the rising FSL and the air region above the free surface [14]. At point (X2) in Figure 8b, which was located within the vegetation patch, the maximum reduction in velocity was observed in the longer vegetation patch (AR < 1), case (1) and case (2) due to continuous resistance provided by the vegetation cylinders in the longitudinal direction. In comparison, this reduction was low in wider vegetation patches (AR > 1), i.e., case (4) and case (5). As a result, a longer vegetation patch benefits the protection of the vegetation structure itself, which helps in reducing bed shear stress inside the vegetation structure. Additionally, the same trend of velocity reduction was observed at points (X3) and (X4) for all the cases in Figure 8c,d. These streamwise velocity profiles in

the vertical direction within and around the vegetation were challenging to observe in the experimental study. This manifests the numerical simulation benefits.

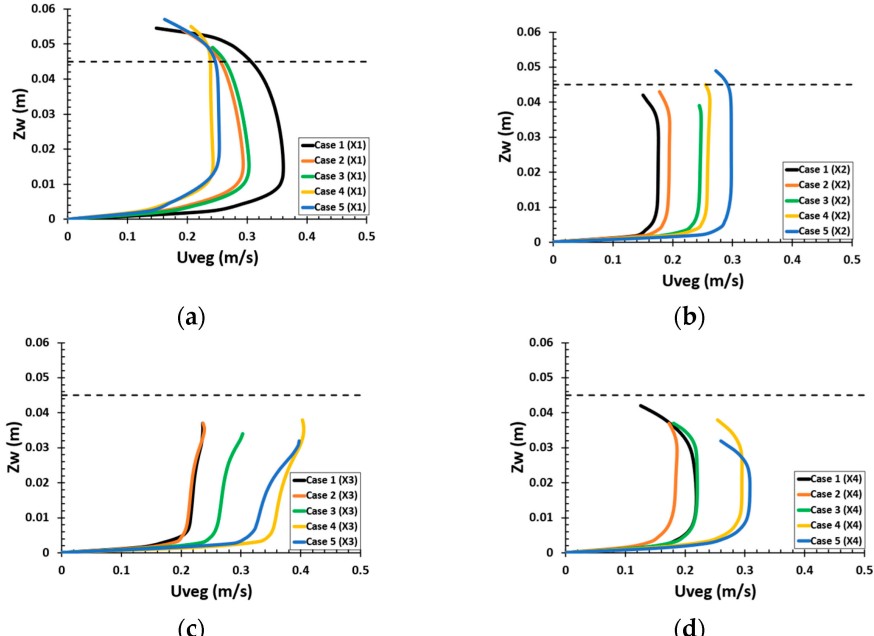

**Figure 8.** Vertical distribution of streamwise velocities in the vegetation region and downstream region at specified locations (**a**) X1; (**b**) X2; (**c**) X3; (**d**) X4. For the locations, see Figure 2.

For all the different AR cases (1–5), the streamwise velocity in the vegetation gap area is shown in Figure 9a–c. These points were considered on one side of the centerline in the gap region because the vegetation is located at the centerline, and conditions are symmetrical on both sides of the centerline. Point (Y1) is in the gap area on the upstream side of the vegetation, and point (Y2) is located at the center of the vegetation gap region, while point (Y3) is located downstream of the vegetation gap region. The *x*-axis depicted the streamwise velocity of the gap region ($U_{Gap}$), whereas the *y*-axis depicted the distance from the bed ($Z_w$) up to the FSL. It is observed in all the AR cases (1–5) that velocity profiles were lower in the upstream gap regions (Y1), marginally higher in the center of the gap region (Y2), increased more in the downstream of the gap region (Y3) and decreased to their minimum near the bed. Drop in velocities located within close vicinity to the bed were attributable to the bed's domain resistance. The streamwise flow velocity was observed to be low in all the aspect ratio cases (1–5) at point (Y1) Figure 9a on the upstream side of the vegetation gap region. This effect of low velocity was due to the sheltering effect of vegetation cylinders and water reflection, i.e., increased height of the FSL, which also affects the gap region. The maximum reduction in streamwise velocity was observed in a wider vegetation patch (AR > 1), i.e., case (5), at point (Y1), due to the increased width of vegetation in the crosswise direction, resulting in a more significant increase in the height of the FSL. The streamwise velocity of case (4) and case (5) is increased as it moves towards point (Y2), Figure 9b. The inflow of water from the vegetation zone into the gap leads to a large-scale eddy between the interface of the gap and the vegetation cylinders. The minimum length of vegetation along the longitudinal direction was also the main factor for increasing the velocity.

The streamwise velocity was further increased at point (Y3), Figure 9c, downstream of the vegetation gap region, and maximum velocity was observed in case (5) due to the fast diffraction of flow at the edges of the vegetation. From the above profiles, it can be shown that case (4) and case (5) provide the highest reduction in velocity upstream of the vegetation gap region, Figure 9a. Simultaneously, these configurations contribute to an increase in velocity downstream of the vegetation gap region Figure 9c, which may lead to

erosion at the vegetation edges. However, in cases (1) and (2), where the vegetation length was greater than the vegetation width, the same pattern of velocities was observed at the upstream, middle, and downstream sides of the vegetation gap region, but the reduction in velocities was less due to the minimum FSL at the upstream side, and the increase in velocities was less due to the increased length of the vegetation patch.

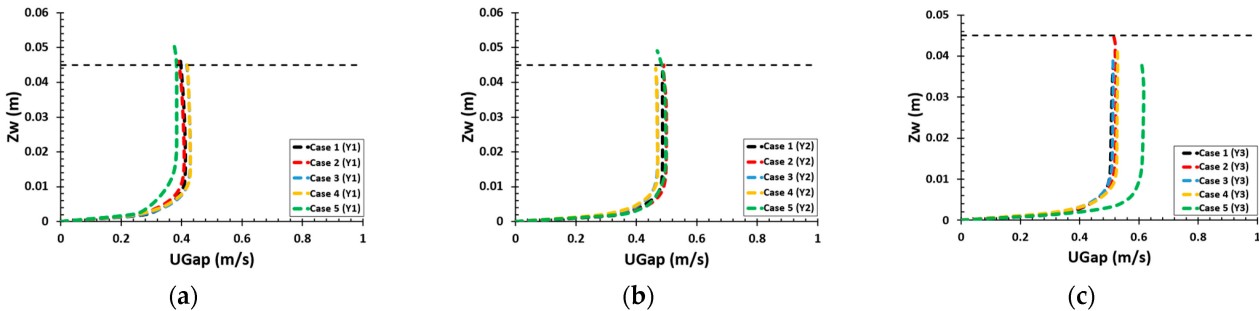

**Figure 9.** Vertical distribution of streamwise velocities in the gap regions at specifies locations (**a**) Y1; (**b**) Y2; (**c**) Y3. For the locations, see Figure 2. The black dashed lined shows the initial FSL.

### 5.2.3. Velocity Profiles along the Longitudinal Sections

The streamwise velocity distribution through the longitudinal sections (LS1 = $y$ = 0.35 m) and (LS2 = $y$ = 0.25 m), i.e., passed from the center of the channel and the gap region, for all the AR cases (1–5) is shown in Figure 10a,b. The dashed boxes depict the vegetation area along the longitudinal direction. In Figure 10a, all the cases followed an irregular pattern of velocities within the vegetation area. The longer the vegetation patch length, i.e., (AR < 1), the maximum irregularity is observed in the velocity profiles. A maximum reduction of 97% and 94.8% in velocities was observed within the vegetation patch in case (1) and case (2), respectively, as compared to the initial velocity without the vegetation. These configurations of vegetation patches could effectively reduce the bed shear stress inside the vegetation patch. When the vegetation patch of (AR ≥ 1) is considered, the velocity reduction in cases 3, 4, and 5 also led to approximately (40.02–53.50%).

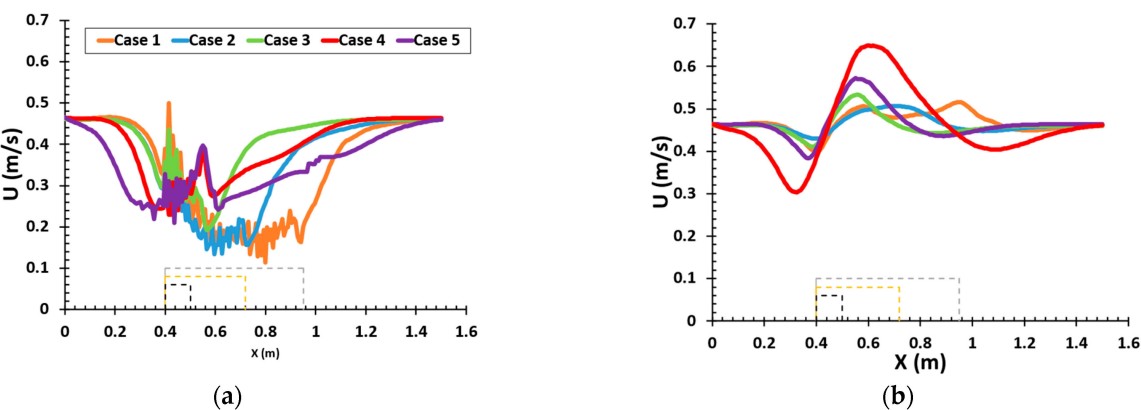

**Figure 10.** Longitudinal distributions of streamwise velocities, (**a**) along the longitudinal section LS1 and (**b**) along the longitudinal section LS2. The dashed lines show the Vegetation length area ($W_x$).

The reduction in streamwise velocities upstream of the vegetation gap region was observed in all the AR cases due to increased FSL, Figure 10b. When the AR is increased, the greater the velocity reduction in the upstream of the gap region (around 11.83–32.7% reduction in velocities compared to initial velocity) was observed in cases 3, 4, and case 5. However, the smaller AR vegetation patch also leads to a reduction in velocity upstream of the vegetation gap area, which in case (1) and case (2) was approximately 7.1% and 14.1%, respectively. Contrarily, with the increase in the AR of vegetation patch, the greater

the velocity increment downstream of the gap region (around 14.26–34.86% increment in velocities compared to initial velocity) was observed in case 3, 4, and case 5. While this increment was only 8.6% and 10.11% in the smaller AR case (1) and case (2), respectively, it suggests that the longer vegetation patch did not substantially contribute to the increase in velocity downstream of the vegetation gap area.

### 5.2.4. Simulated Contour Plot Distribution of Streamwise Velocity

The contour plot distribution of streamwise velocity on the horizontal *x-y* plane for all the cases is shown in Figure 11a–e. One horizontal surface was examined at depth $z = 3.5$ cm (below the free surface) to better understand the flow behavior. When the flow bleeds through the vegetation region, the reduction in stream wise velocities was observed at the front of the vegetation in all the cases (1–5). The lowest reduction was observed in longer vegetation patch configuration, i.e., case (1), case (2) and case (3), Figure 11a–c, whereas the highest reduction was seen in the wider patch configuration, i.e., case (4) and case (5), Figure 11d,e. This was due to the presence of the maximum number of vegetation cylinders in the path of flow. Regarding the inside of the vegetation region, the trend of reduction in velocities were different. Due to the continuous resistance provided by the vegetation cylinders, the highest reduction in velocity was observed in longer vegetation patches (AR < 1), considering the inside region of the vegetation patch. However, wider vegetation patches (AR > 1) also contribute to velocity reduction, but it produces a large, sheltered zone (area of low velocity), Figure 11d,e, downstream of the vegetation patch compared to the longer vegetation patches (AR < 1). Therefore, considering the tsunami mitigation capability point of view, the findings illustrate the significance of a wider vegetation patch, which could significantly increase the maximum velocity reduction area just behind the vegetation patch compared to the longer vegetation patch.

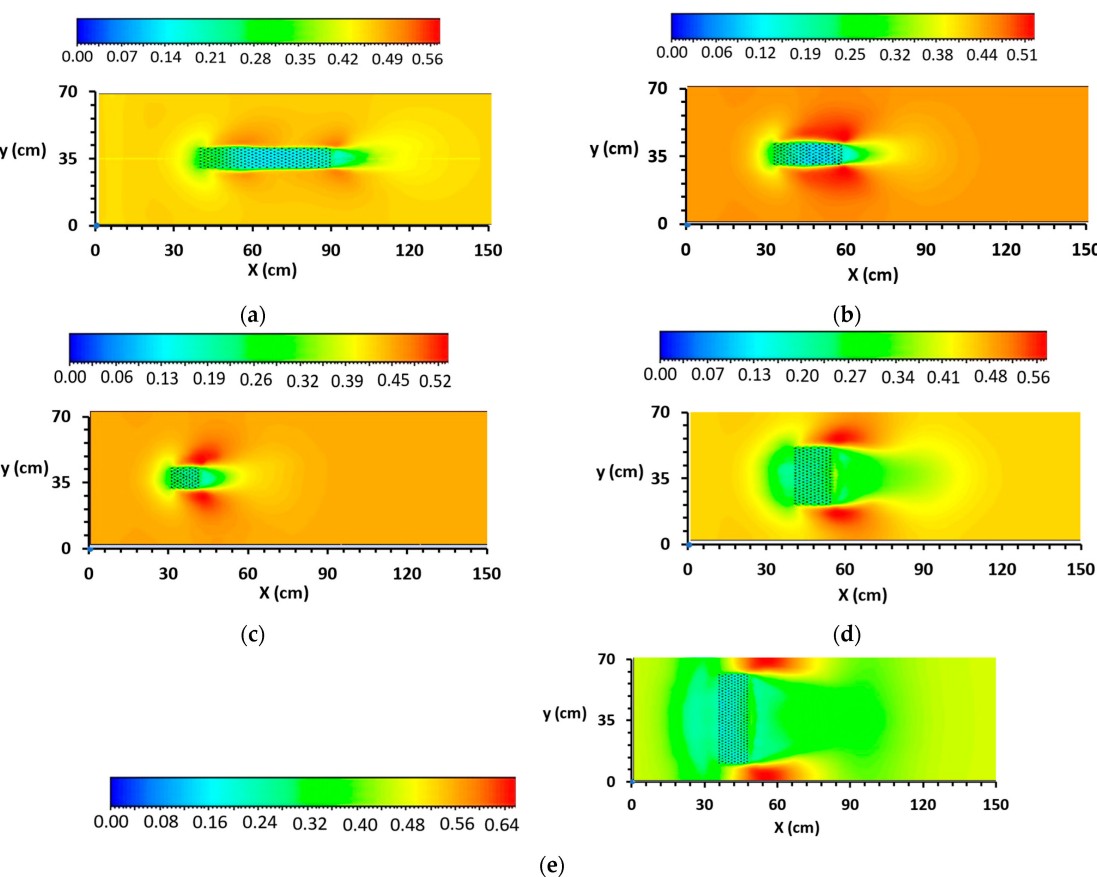

**Figure 11.** Contour plot distribution of streamwise velocity along the horizontal plane: (**a**) Case 1, (**b**) Case 2, (**c**) Case 3, (**d**) Case 4, (**e**) Case 5.

The sheltering effect of the longer vegetation patch, on the other hand, causes the velocity to decelerate in the nearby gap areas when compared to the wider vegetation patch. Therefore, the wider vegetation patches (AR > 1) contribute to generating a small zone of high-intensity velocity magnitude in the vegetation gap region, and the longer vegetation patches (AR < 1) lead to the formation of a large zone of low-intensity velocity magnitude. Therefore, an increased AR of the vegetation patch has a negative impact on the downstream gap region due to increased velocity magnitudes at the vegetation patch edges compared to the longer vegetation patch.

### 5.2.5. Flow Movement of Water and Air Phase

The schematic detail of the flow behavior between the air and water phase and generation of vortices at specified sections (CS1) and (CS2) for the only three AR cases (1, 3 and 5) are shown in Figure 12a–f. The *y*-axis represents the channel's crosswise direction, i.e., the channel's width and the *z*-axis represents the channel's total depth, including both the air and water region. Due to the slip wall presence, only the vegetated region and bed of the channel domain were accountable for the generation of vortices. The presence of vegetated and gap regions contributes to an inflection point along both the crosswise sections (CS1) and (CS2) in the streamwise velocity vectors. In section (CS1) near the vegetation region (water phase), i.e., the interface between the vegetated and gap region, flow divergence; that is, the movement of flow in a lateral direction, occurred in the regions of vegetation area towards the gap region, and, as a result, the downward movement of water flow occurred within the vegetation region, Figure 12a,c,e. Contrarily, flow convergence in section (CS2); that is, the movement of water flow, took place in the reverse direction just at the downstream side of the vegetation patch in a lateral direction, as the flow behind the vegetation again collided and, as a result, an upward movement of flow occurred in front of the downstream side of the vegetation patch Figure 12b,d,f.

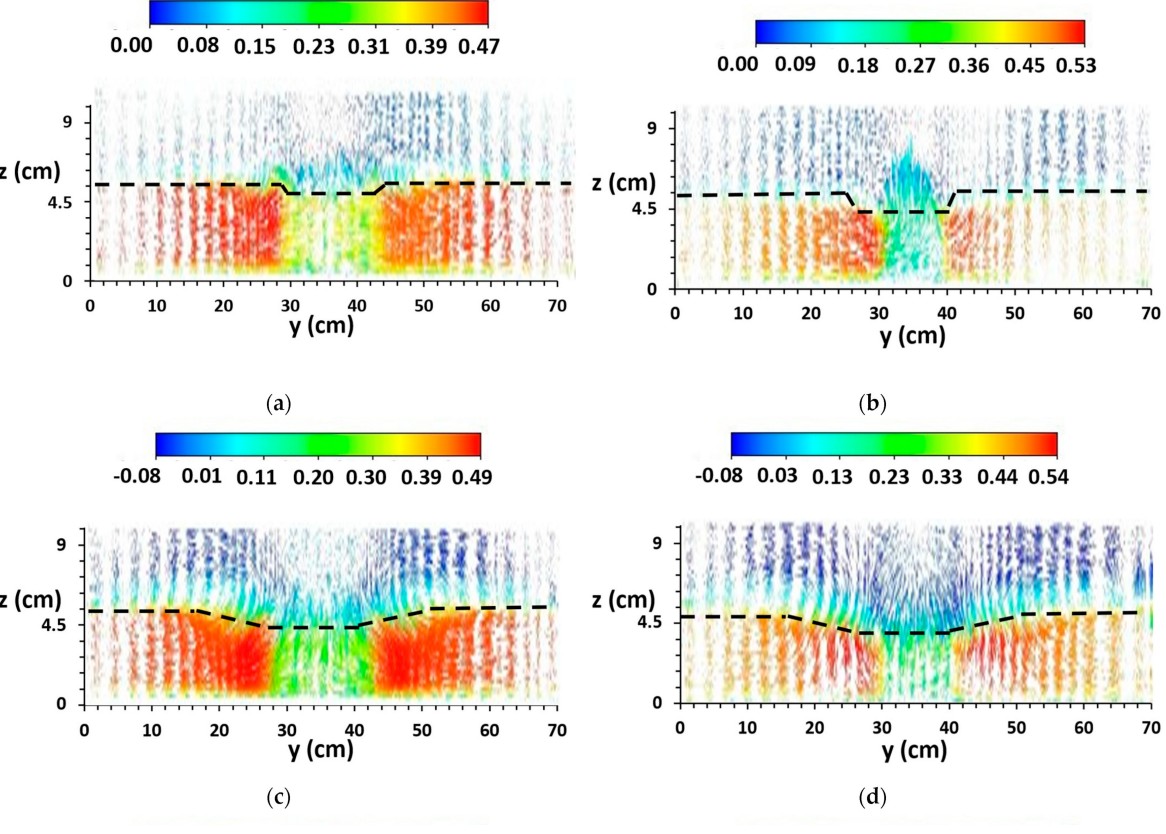

**Figure 12.** *Cont.*

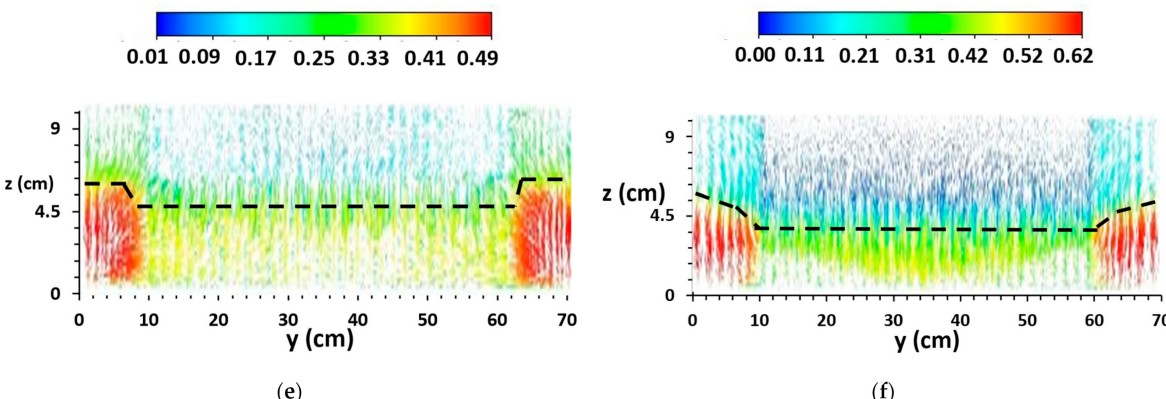

(e)

(f)

**Figure 12.** Streamwise velocity vectors (m/s) along the specified cross sections CS1 and CS2 (**a**,**b**) Case 1; (**c**,**d**) Case 3; (**e**,**f**) Case 5. The black dashed lined shows the FSL (Free Surface Level).

Owing to the combination of the air and water layers, the inflection point in the streamwise velocities was seen near the free surface level in all three AR cases. The greater velocity difference of the two phases, i.e., air and water, contribute to a generation of strong vertical clockwise and counterclockwise vortices in the vertical direction Figure 13a,b. The combination of low-density air–liquid and high-density water–liquid was accountable for these vortices. However, these vortices are more prominent when the gap region is larger, i.e., case (1) and case (3). Above the FSL, a downward air movement occurs in the region above the vegetation area and the regions immediately behind the vegetation side. According to Rashedunnabi and Tanaka [14], the air movement that occurs in the downward direction in a vegetated open channel flow is the main reason for the air entrainment in the vegetation patch at the downstream side, which can be seen in the present numerical study.

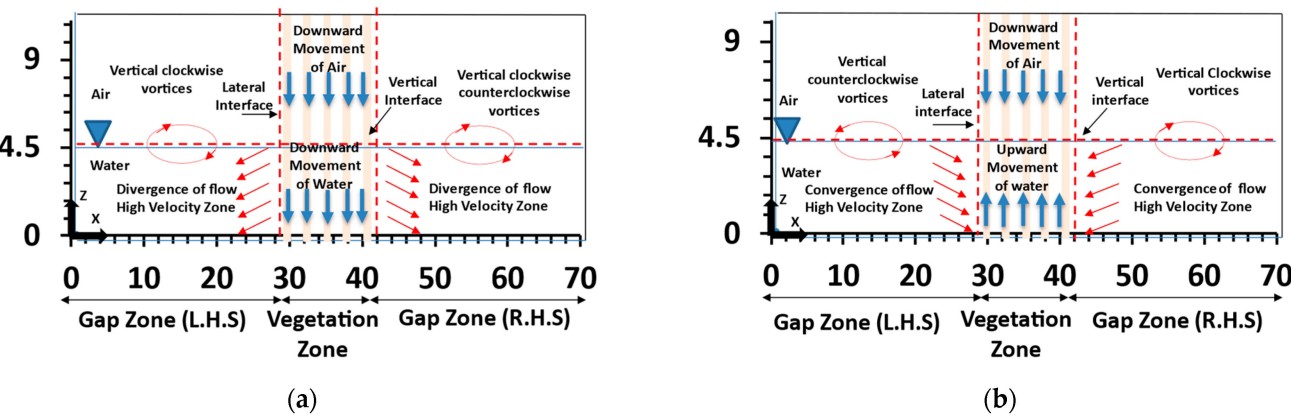

(**a**)

(**b**)

**Figure 13.** Schematic diagram of general vortex behavior in an emergent vegetated channel of air and water behavior along with the cross-sections (**a**) CS1; (**b**) CS2 for all the cases.

### 5.2.6. Turbulent Intensity Distribution of Air–Water Phase

When the flow bleeds through the vegetation area, the flow becomes unstable in that region. The flow instability in the vegetation region leads to the turbulence level. In addition, the air region above the water region also affects the flow phenomenon in terms of air entrainment into the water region. Therefore, the estimation of the turbulence level of both the air and water regions is very important and can be easily understood by using VOF modeling. The turbulent intensity ($I = \sqrt{\overline{u'^2}} \big/ u_{avg}$) in the form of contour plots for the three AR cases (1, 3 and 5) only at the specified section (CS1) is shown in Figure 14a–c. On the x-axis is the width of the channel, and on the z-axis is the elevation, which includes both

air and water regions. Figure 14a–c shows that the maximum water turbulence intensity below the free surface was observed both within the vegetation and near the bed of the channel domain. This was due to the vegetation cylinders resistance in the path of the water flow and the continuous resistance provided by the bed of the domain. Raupach et al. [49] and Ricardo et al. [50] found that the production of vortices within the vegetation leads (as seen in Figure 12a,c,e) to the dissipation of the turbulent energy. A considerable quantity of turbulence was detected near the vegetation edge (i.e., the boundary between the gap region and the vegetation region), which resulted from the rapid water moving towards the gap area (discussed in Section 5.2.5). The gap regions exhibited little resistance; thus, very little turbulence was recorded there. Therefore, the maximum turbulent intensity in the vegetation region showed that vegetation cylinders provide significant resistance in the path of the flow.

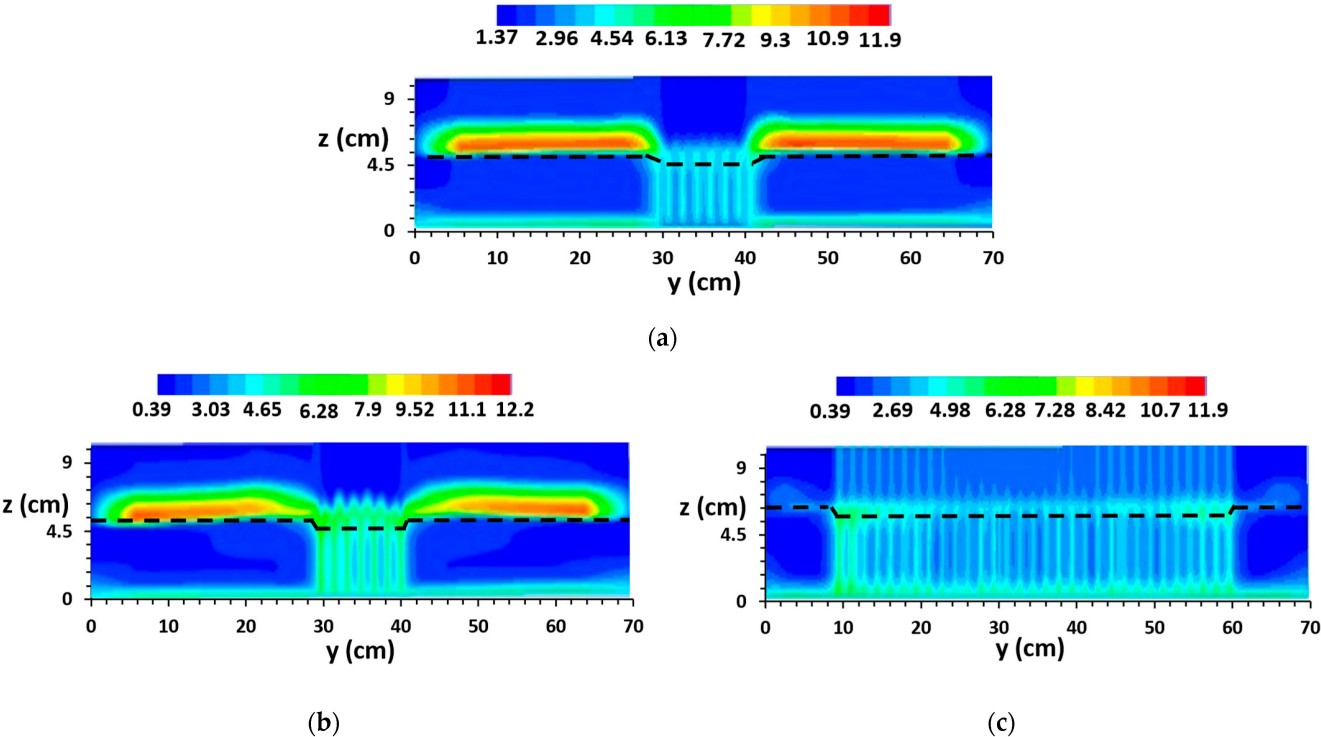

**Figure 14.** Turbulent intensities (%) along the cross-section CS1 (**a**) Case 1; (**b**) Case 3 and (**c**) Case 5.

Savelsberg and Van de water [51] found that turbulence of air at the free surface indicates air entrainment into the water region. Above the FSL, in the gap region, strong air turbulent intensity was observed in case (1), 11.9%, and case (3), 12.1%, indicating strong air entrainment over this region, Figure 14a,b. In the gap region of the maximum AR case (5), less than 3% of air turbulent intensity was gained over the free surface level due to constriction of the gap area. while a significant increase in air turbulent intensity above the vegetation region was found in case (5), i.e., wider vegetation patch, Figure 14c. A study by Cain [52] revealed that air entrainment occurred in a turbulent flow due to irregular vortices' formation. In the present study, these vortices were formed at the free surface level, as discussed in the above section. In addition, Chanson [53] showed that if the turbulent properties are high enough to transcend both gravity and surface tension impact, the air bubbles can be entrained. Therefore, a significant amount of air bubbles forming in a maximum aspect ratio can result from the higher turbulent intensity of air above the vegetation cylinders. Previous studies have also shown that higher generation of turbulent structures can promote the flow of air bubbles into the vegetation region [54,55], which can play an important role in energy dissipation [14,56]. Therefore, the maximum

air intensity above the wider vegetation patch region (AR > 1) can result in greater energy reduction relative to that of longer vegetation patch configuration (AR < 1).

## 6. Conclusions

In this research, a three-dimensional numerical model was established using ANSYS to investigate inland tsunami flow in a vegetated open channel. The two-phase volume of fluid (VOF) model was used to track the free surface combined with the Reynolds stress model (RSM). The accuracy of the free surface was checked using a variety of meshing methods; however, multizone meshing was the most effective in correctly tracking the free surface with the smallest amount of diffusion, compared to the tetrahedral meshing. The present numerical model has successfully captured reasonable outcomes and has better correspondence with the experimental results in terms of free surface and velocity profiles.

The numerical model was then utilized to investigate the 3D flow behavior through the longer (AR < 1) and wider (AR > 1) vegetation patches to compare the flow behavior of each type under subcritical flow conditions depending on the Aspect Ratio (AR = vegetation patch width/length). The results showed that the maximum free surface level (FSL) at the front and minimum FSL behind the vegetation increases with the increase in AR > 1 (in the case of the wider vegetation patch), which consequently results in the generation of large size vortices. On the other hand, with the decrease in AR $\leq$ 1 (in the case of the longer vegetation patch), less FSL is observed at the front of the vegetation, which consequently results in the generation of small size vortices. Considering the tsunami mitigation point of view, the wider vegetation patch configurations (AR > 1) give favorable conditions in terms of the large area of low velocity directly downstream side of the vegetation patch, whereas it has a negative impact on the adjacent downstream gap region due to increased high velocity (around 14.3–34.9% increment in velocities compared to initial velocity), which would increase bed shear stress at the edges of the vegetation. The longer vegetation patch configurations (with AR < 1) gives favorable conditions due to maximum velocity reduction within the vegetation patch (around 94.8–97.14% decrement in velocities), which would decrease bed shear stress and protect the vegetation structures itself. However, it has a disadvantage in the downstream vegetation region due to the generation of a small, sheltered area of low velocity. In addition, the present numerical model successfully captured the flow movement of water and air region around the vegetation patches. The turbulent intensity distribution of air and water region showed that the turbulence of air at the free surface suggests strong air entrainment into the water region, which can result in greater energy reduction. The maximum air intensity above the wider vegetation patch region (AR > 1) was observed compared to the longer vegetation patch configuration (AR < 1).

These results are beneficial and provide basic information for considering the suitable design of finite length emergent vegetation based on the Aspect Ratio (AR). Therefore, in the future, more computational research should be conducted to analyze the flow properties against the finite length emergent vegetation, with further varying configurations and with some angled gaps to overcome the high-velocity zone at the edges of the vegetation patch (observed in the wider patch configuration) and other ground conditions.

**Author Contributions:** Conceptualization, A. and N.T.; methodology, A.; software, A.; validation, A.; writing—original draft preparation, A.; review, N.T.; supervision, N.T. All authors have read and agreed to the published version of the manuscript.

**Funding:** This research received no external funding.

**Institutional Review Board Statement:** Not applicable.

**Informed Consent Statement:** Not applicable.

**Data Availability Statement:** Not applicable.

**Acknowledgments:** The authors acknowledge the support of MEXT scholarship (A from the Japanese Ministry of Education, Culture, Sports, Science, and Technology (Monbukagakusho).

**Conflicts of Interest:** The authors declare no conflict of interest.

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
