# Peer review of "Numerical Investigation of 3D Flow Properties around Finite Emergent Vegetation by Using the Two-Phase Volume of Fluid (VOF) Modeling Technique"

_fluids, doi:10.3390/fluids7050175_

Round 1

Reviewer 1 Report

Interesting study. I have not seen flow-vegetation interaction based on aspect ratio analysis. My only concern is the authors have not report any sort of uncertainty in the entire paper. All these measurements (e.g. velocity values,...) need uncertainty. 

Author Response

Author’s Response to the Review Comments

Manuscript No. 1707924

Title: Numerical investigation of 3D flow properties around finite length emergent vegetation by

using the two-phase volume of fluid (VOF) modeling technique

Corresponding Author: Prof. Dr. Norio Tanaka

All Authors: Amina Amina, Norio Tanaka*

We appreciate the opportunity to submit a revised version of our paper, so thank you very much. We are grateful to the editor and reviewers for their time and effort in evaluating this article. All the reviewers' comments have been addressed thoroughly in this file. For the sake of clarity, we've copied the reviewers' comments and responded to each one, along with the section and line number from the revised manuscript. It is our belief that we have addressed all the issues and that the updated version will meet the journal's requirements for publication.

Reviewer 1:

Comment:

Interesting study. I have not seen flow-vegetation interaction based on aspect ratio analysis. My only concern is the authors have not report any sort of uncertainty in the entire paper. All these measurements (e.g. velocity values,...) need uncertainty. 

Response:

Thank you so much for your kind review. Before starting the postprocessing process of our analysis, we need to build a strong mesh around the numerical domain to get accurate results like the free surface variations and velocity changes around the vegetation patch and minimize the uncertainty in the results. Therefore, we performed the grid independency test to achieve the accurate results. A detailed review on the grid independency test is now added in revised manuscript.

Changes have been made Line 282-296

We're grateful to all of the reviewers who took the time to provide their thoughts and suggestions. We have done our utmost effort to address all of the issues in the revised manuscript and believe that the updated version can meets the journal's publishing criteria.

The velocity patterns are nearly impossible to capture through an experimental investigation due to the limitations of instrumental use such as laser sheet in a Particle Image Velocity Meter (PIV). Because it was unable to use in the vegetation zone due to the presence of vegetation cylinders. Also, velocity sensor in the Electromagnetic Flow Meter (EFM) was unable to use because it disturbed the flow near the vegetation region and the bed region.

 , where   represents the initial flow velocity, g is the gravitational constant and  is the initial water depth)

Reviewer 2 Report

The paper presented the numerical simulations for 3D flow properties around finite length emergent vegetation. The results are helpful to design vegetation arrangement for tsunami mitigation. Some parts of the paper need to be improved or clarified before the paper can be accepted. The followings are some concerns. Please check the entire manuscript carefully.

  • Lines 69 to 73: rewrite the sentence;
  • Italic and Non-Italic symbols are different. The authors should ensure the symbols are coincident in the manuscript.
  • Table 1 Line and text are overlapped.
  • 2.1. and 4.2.2. Preprocessing
  • 6b, the continuous line should be the numerical results, and the scatter points should be the experimental results.
  • Definition of Froude number
  • Definition of S/d and what is the corresponding density? which is the parameter commonly used in previous literatures.
  • Figure 7 Colorbar for (e)(f)(g)(h)
  • Figure 12 unit for the vector
  • Figure 14 unit for the figure. How do you define turbulent intensities? Does it include density? Please define it in the manuscript.

Author Response

Author’s Response to the Review Comments

Manuscript No. 1707924

Title: Numerical investigation of 3D flow properties around finite length emergent vegetation by

using the two-phase volume of fluid (VOF) modeling technique

Corresponding Author: Prof. Dr. Norio Tanaka

All Authors: Amina Amina, Norio Tanaka*

We appreciate the opportunity to submit a revised version of our paper, so thank you very much. We are grateful to the editor and reviewers for their time and effort in evaluating this article. All the reviewers' comments have been addressed thoroughly in this file. For the sake of clarity, we've copied the reviewers' comments and responded to each one, along with the section and line number from the revised manuscript. It is our belief that we have addressed all the issues and that the updated version will meet the journal's requirements for publication.

Reviewer 2:

Comments

  1. Lines 69 to 73: rewrite the sentence;
  2. Italic and Non-Italic symbols are different. The authors should ensure the symbols are coincident in the manuscript.
  3. Table 1 Line and text are overlapped.
  4. 1. and 4.2.2. Preprocessing
  5. 6b, the continuous line should be the numerical results, and the scatter points should be the experimental results.
  6. Definition of Froude number
  7. Definition of S/d and what is the corresponding density? which is the parameter commonly used in previous literatures.
  8. Figure 7 Color bar for (e)(f)(g)(h)
  9. Figure 12 unit for the vector
  10. Figure 14 unit for the figure. How do you define turbulent intensities? Does it include density? Please define it in the manuscript.

Response:

Thankyou for your kind comments. Now we have incorporated all the comments in the revised version of manuscript.

  1. Now we have rewritten the sentence. Line 73-76
  2. Now the symbols are corrected in the revised version of manuscript.
  3. Now the issue has been resolved.
  4. Now we have corrected the word Line 329
  5. The graphs have been changed in the revised manuscript (Figure 6).
  6. Definition of Froude number has been added. Line 208-209
  7. The vegetation is defined by S/d number. Previous research defined its values and based on that value they classified the density of vegetation. Details are given in the revised manuscript. Line 179-184
  8. The color bar is common for figure 7
  9. Noe the unit has been mentioned in the revised text. Line 736
  10. The formula has been given in the revised manuscript and intensities are presented in percentage and now mentioned in the revised text. Line 776, 882.

We're grateful to all of the reviewers who took the time to provide their thoughts and suggestions. We have done our utmost effort to address all of the issues in the revised manuscript and believe that the updated version can meets the journal's publishing criteria.

Reviewer 3 Report

This paper studies the free surface level variations around finite length vegetation by using a RANS model with VOF and Reynolds stress model. Although the RANS model is not new, this work is still “important in constructing vegetation layouts based on the AR of the vegetation for tsunami mitigation”. The English in this paper is good, despite some minor issues, and the results presented are very interesting. In my opinion, this paper should be accepted, but I have a few concerns that must be addressed before its publication:

  1. This sentence in the abstract is confusing to me: “The results showed that the generation of large size vortices was predominated in wider vegetation patches (AR>1) due to increasing and decreasing the FSL at the front and back of the vegetation as compared to longer vegetation patches (AR≤ 1). “ I would move “due to increasing and decreasing the FSL at the front and back of the vegetation” to the end of this sentence.
  2. In the literature review, the authors thoroughly discussed the prior studies on “density of vegetation, species, dimensions, alignments, and scale of vegetations”. However, references are only provided for the factors of density and scale. The author could consider adding reference to studies about the other factors like species (e.g., Tanaka (2009) and Tanaka et al. (2011)), dimensions (e.g., Ismail et al. (2012)), etc.
  3. The literature review about finite vegetation coverage mostly focuses on open gaps, and this manuscript’s study focuses on a single patch. There are considerable studies about discontinuous vegetation coverage such as patch arrays (e.g., Vandenbruwaene et al. (2011), Irish et al. (2014)), which is a common distribution pattern in nature. The authors can consider adding a review on such studies about the flow between patches.
  4. The flow condition in this study is in a single inundation direction, while in reality, tsunami usually includes inundation (e.g., run-up) and withdrawal, both of which could cause tremendous flux (e.g., Carrier et at. (2003)). In addition, if discontinuous vegetation is present, the hydraulic loads could also vary significantly in different areas (e.g., Yang et al. 2017, Zainali et al. 2018). The authors could consider discussing the similarity and differences between their results and the other studies of bi-directional flow as well as the spatial varying loads between patches
  5. Line 87, in my opinion, the VOF has not been used to this problem because of the complexities of the flow around the cylinders. Have the authors validated the model against available observational data or experimental results? As mentioned later in this paper, the authors used the RANS model and there should be lots of benchmark cases to validate the numerical model.
  6. Line 235, since the authors mentioned that “Simulation precision, convergence and speed are all influenced by mesh size and density”, I would recommend the authors add a “grid independence test” for this application. Will the other experimental results be any different when using more coarser mesh or finer mesh? In addition, theoretically tetrahedral meshing should not generate incorrect results. Can the authors explain why the diffusion rate is so high?
  7. Line 295, SIMPLE algorithm an acronym for Semi-Implicit Method for Pressure Linked Equations. It is NOT a “simple algorithm”. The SIMPLE algorithm should be used for steady flow problems and PISO should be used for unsteady flow. The authors should also add the numerical schemes used in this paper. Which numerical scheme is used for time integration, convection, and diffusion?
  8. For cases 4 and 5, the boundary effect should not be ignored. Why did the authors place the boundary so close to the patches? Are the simulation results validated by the experimental results? In my opinion when the patches are close to the boundary, inevitably the flow conditions change. Are the different flow properties due to the patches or due to the boundary effect?

Several minor concerns:

  1. In the title the authors mentioned “finite length emergent vegetation”, but the word “length” is confusing because the “height” of the vegetation can be also called “length”.
  2. Line 64, AR should be defined earlier before using it. 
  3. Line 110, what are partial_t and partial_x_t in Eq. (1)? 
  4. Line 131, the points in u’_i and u’_j are missing on my PDF reader.
  5. The mesh domain in Fig. 3 is not clear.
  6. The titles for 4.2.1 and 4.2.2 are all “preprocessing”
  7. Fig. 6 (b), why are there two red dots at the bottom of this figure?

Suggested references:

Tanaka, N., 2009. Vegetation bioshields for tsunami mitigation: review of effectiveness, limitations, construction, and sustainable management. Landscape and Ecological Engineering 5, 71–79. 

Tanaka, N., Jinadasa, K., Mowjood, M.I.M., Fasly, M.S.M., 2011. Coastal vegetation planting projects for tsunami disaster mitigation: effectiveness evaluation of new establishments. Landscape and ecological engineering 7, 127–135. 

Ismail, H., Abd Wahab, A., and Alias, N., 2012. “Determination of man- grove forest performance in reducing tsunami run-up using physical models.” Nat. Hazards, 63(2), 1–25. 

Vandenbruwaene, W., et al., 2011,. “Flow interaction with dynamic vegeta- tion patches: Implications for biogeomorphic evolution of a tidal land- scape.” J. Geophys. Res. Earth Surf., 116(F1), F01008 

Irish, J. L., Weiss, R., Yang, Y., Song, Y. K., Zainali, A., and Marivela- Colmenarejo, R., 2014. “Laboratory experiments of tsunami run-up and withdrawal in patchy coastal forest on a steep beach.” Nat. Hazards, 74(3), 1–17. 

Carrier G, Wu TT, Yeh H, 2003, “Tsunami run-up and draw-down on a plane beach.” J Fluid Mech 475:79–99 

Yang, Y., Irish, J.L., Weiss, R., 2017. “Impact of patchy vegetation on tsunami dynamics”. J. Waterw. Port C.-ASCE 143 (4), 04017005.

Zainali, A., Marivela, R., Weiss, R., Yang, Y., & Irish, J. L., 2018, “Numerical simulation of nonlinear long waves in the presence of discontinuous coastal vegetation.” Marine Geology, 396, 142-149.

Author Response

Author’s Response to the Review Comments

Manuscript No. 1707924

Title: Numerical investigation of 3D flow properties around finite length emergent vegetation by

using the two-phase volume of fluid (VOF) modeling technique

Corresponding Author: Prof. Dr. Norio Tanaka

All Authors: Amina Amina, Norio Tanaka*

We appreciate the opportunity to submit a revised version of our paper, so thank you very much. We are grateful to the editor and reviewers for their time and effort in evaluating this article. All the reviewers' comments have been addressed thoroughly in this file. For the sake of clarity, we've copied the reviewers' comments and responded to each one, along with the section and line number from the revised manuscript. It is our belief that we have addressed all the issues and that the updated version will meet the journal's requirements for publication.

Reviewer 3

Comment 1:

  1. This sentence in the abstract is confusing to me: “The results showed that the generation of large size vortices was predominated in wider vegetation patches (AR>1) due to increasing and decreasing the FSL at the front and back of the vegetation as compared to longer vegetation patches (AR≤ 1). “ I would move “due to increasing and decreasing the FSL at the front and back of the vegetation” to the end of this sentence.

Response:

Thank you so much for highlighting this issue. The resistance offered by wider patches were more as compared to longer vegetation patches against the approaching flow, that’s why large size vortices were predominated in the wider vegetation patches and hence rise and fall of Free surface Level (FSL) at the upstream and downstream side of vegetation was more. To avoid confusion, we have again rewritten the sentence for better clarification.  

Changes have been made Line 20-21

Comment 2:

  1. In the literature review, the authors thoroughly discussed the prior studies on “density of vegetation, species, dimensions, alignments, and scale of vegetations”. However, references are only provided for the factors of density and scale. The author could consider adding reference to studies about the other factors like species (e.g., Tanaka (2009) and Tanaka et al. (2011)), dimensions (e.g., Ismail et al. (2012)), etc.

Response:

Thank you for providing the literature regarding vegetation properties. Now we have incorporated it in the literature review.

Changes have been made Line 47

Comment 3:

  1. The literature review about finite vegetation coverage mostly focuses on open gaps, and this manuscript’s study focuses on a single patch. There are considerable studies about discontinuous vegetation coverage such as patch arrays (e.g., Vandenbruwaene et al. (2011), Irish et al. (2014)), which is a common distribution pattern in nature. The authors can consider adding a review on such studies about the flow between patches.

Response:

Thank you for your recommendation about adding a review about discontinuous patches in literature review. Now we have added it in the literature review.

Changes have been made Line 60-61

Comment 4:

  1. The flow condition in this study is in a single inundation direction, while in reality, tsunami usually includes inundation (e.g., run-up) and withdrawal, both of which could cause tremendous flux (e.g., Carrier et at. (2003)). In addition, if discontinuous vegetation is present, the hydraulic loads could also vary significantly in different areas (e.g., Yang et al. 2017, Zain ali et al. 2018). The authors could consider discussing the similarity and differences between their results and the other studies of bi-directional flow as well as the spatial varying loads between patches.

Response:

Thank you for your valuable comment. The reviewer is right that the tsunami, which is a series of waves, continues to the shore and runs up inland with a huge discharge. Whereas when the tsunami inundation enters in the inland regions, it does not contain any waves and propagates into the inland region with a long period. As a result, the tsunami inundation can be represented as a quasi-steady flow. Many earlier researchers considered the flow in the inland area surrounding an inland vegetative forest in a steady and subcritical condition when modeling the tsunami flow. Thus, the flow parameters in this research were defined using Froude and initial water depth similarity, which assumes a subcritical and constant tsunami inland flow.

Change have been made Line 152-159

Comment 5:

  1. Line 87, in my opinion, the VOF has not been used to this problem because of the complexities of the flow around the cylinders. Have the authors validated the model against available observational data or experimental results? As mentioned later in this paper, the authors used the RANS model and there should be lots of benchmark cases to validate the numerical model.

Response:

Thank you for your kind comment. Previously Debnath (2008) used VOF model along with RANS model in vegetated open channel flow in which submerged vegetation was used. In our study firstly we validated our numerical model with the conducted experimental data and numerical model results are consistent with the findings of the experimental data results.

Reference

Debnath, K.; Bhattacharya, A. K.; Mahato, B.; Chakrabarti, A.  Volume of fluid model for numerical simulation of vegetated flows. ISH J. Hydraul. Eng 2008, 14, 72–87.

Comment 6:

  1. Line 235, since the authors mentioned that “Simulation precision, convergence and speed are all influenced by mesh size and density”, I would recommend the authors add a “grid independence test” for this application. Will the other experimental results be any different when using more coarser mesh or finer mesh? In addition, theoretically tetrahedral meshing should not generate incorrect results. Can the authors explain why the diffusion rate is so high?

Response:

Thankyou for your recommendation. Now we have added the mesh independence test and elaborated the results by changing the mesh size of the numerical domain. Also, we have added the details of why diffusion rate is so high in tetrahedral meshing.

Change have been made Line 285-299, Line 270-274

Comment 7:

  1. Line 295, SIMPLE algorithm an acronym for Semi-Implicit Method for Pressure Linked Equations. It is NOT a “simple algorithm”. The SIMPLE algorithm should be used for steady flow problems and PISO should be used for unsteady flow. The authors should also add the numerical schemes used in this paper. Which numerical scheme is used for time integration, convection, and diffusion?

Response:

Thank you for your correction. In present study, the steady case was considered, and SIMPLE algorithm was used. Now we have added the description of the schemes used in present study.

Change have been made Line 359-342

Comment 8:

  1. For cases 4 and 5, the boundary effect should not be ignored. Why did the authors place the boundary so close to the patches? Are the simulation results validated by the experimental results? In my opinion when the patches are close to the boundary, inevitably the flow conditions change. Are the different flow properties due to the patches or due to the boundary effect?

Response:

Thank you for your valuable comment. In present study, the boundary condition assigned to the side walls as Slip-wall, which means there is no effect of the wall on the flow properties means no friction from the wall. Therefore, in case 4 and case 5 i.e., wider patch configurations, there is no effect of the wall on the flow properties.

Several minor concerns:

  1. In the title the authors mentioned “finite length emergent vegetation”, but the word “length” is confusing because the “height” of the vegetation can be also called “length”.
  2. Line 64, AR should be defined earlier before using it. 
  3. Line 110, what are partial_t and partial_x_t in Eq. (1)? 
  4. Line 131, the points in u’_i and u’_j are missing on my PDF reader.
  5. The mesh domain in Fig. 3 is not clear.
  6. The titles for 4.2.1 and 4.2.2 are all “preprocessing”
  7. Fig. 6 (b), why are there two red dots at the bottom of this figure?

Response:

Thank you for highlighting minor issues. Now we have solved all the minor issues in the revised manuscript.

We're grateful to all of the reviewers who took the time to provide their thoughts and suggestions. We have done our utmost effort to address all of the issues in the revised manuscript and believe that the updated version can meets the journal's publishing criteria.

Round 2

Reviewer 3 Report

The authors have addressed most of my comments in a good manner. However, there are still two minor issues:

  1. Although "slip wall" condition is used, the boundary effect is not discussed in this paper appropriately. The authors should either (1) add more experiments having boundary far away from the patches (2) or discuss the boundary effect in the experiments.
  2. The authors did not mention two papers in my first review. 

[1] Yang, Y., Irish, J.L., Weiss, R., 2017. “Impact of patchy vegetation on tsunami dynamics”. J. Waterw. Port C.-ASCE 143 (4), 04017005.

[2] Zainali, A., Marivela, R., Weiss, R., Yang, Y., & Irish, J. L., 2018, “Numerical simulation of nonlinear long waves in the presence of discontinuous coastal vegetation.” Marine Geology, 396, 142-149.

Author Response

Author’s Response to the Review Comments

Manuscript No. 1707924

Title: Numerical investigation of 3D flow properties around finite length emergent vegetation by

using the two-phase volume of fluid (VOF) modeling technique

Corresponding Author: Prof. Dr. Norio Tanaka

All Authors: Amina Amina, Norio Tanaka*

Reviewer 3

Comment 1:

  1. Although "slip wall" condition is used, the boundary effect is not discussed in this paper appropriately. The authors should either (1) add more experiments having boundary far away from the patches (2) or discuss the boundary effect in the experiments.

Response:

Thank you for your valuable comment. Now the details of the boundary effect in the experimental trail have been added.

The boundary effect, also known as the side wall effect, has an influence on the channel flow structure when the blockage ratio, which measures how wide the vegetation patch front is in relation to how wide the channel itself is, exceeds a certain threshold value. According to previous studies, an utterly trivial impact of the side walls on the drag of a flat plate occurs when the obstruction ratio is equal to 5–6 percent [46]. To determine how the drag force decreases with increasing obstruction ratio (up to 40 percent), Okajima et al. [47] performed an experiment around a cylinder shaped like a rectangle, in which the flow rate was constant. They reported that the drag forces initially drop (obstruction ratio of 9 to 10 percent), and then rise as flow obstruction increases. The vegetation model (VM) studied in the present research has the capacity to enable water to flow through it, and this is mostly based on the density. As a result, in order to compute the blockage ratio, the front width (Wc) of the vegetation was estimated by multiplying the number of cylinders in each of the first two rows due to its staggered arrangement by the cylinder diameter. Then, the front width of the vegetation patch is divided by the channel's width. The considered vegetation model for experimental trial had a blockage ratio up to approximately 13%, which was under critical limit. Furthermore, it was also observed during the testing that no waves were reflected from the channel's side wall into the area under investigation during the experimental trials. As a result, it was assumed that the boundary effect had no influence on the results and the effect of the restriction was overlooked.

References:

  • Okamoto, S.; Okamoto, T. Theoretical study of blockage effect of wind-tunnel on wake of two-dimensional flat plate normal to plane wall. Japan Soc. Aero. Space Sci. 1984, 27, 134–144.
  • Okajima, A.; Yi, D.; Kimura, S.; Kiwata T (1997) The blockage effects for an oscillating rectangular cylinder at moderate Reynolds number. J. Wind Eng. Ind. Aerodyn. 1997 69–71, 997–1011.

Changes have been done Line 374-393

Comment 2:

  1. The authors did not mention two papers in my first review. 

Response:

Sorry for the inconvenience. Now the references have been added in the revised manuscript.

Changes have been done Line 150-151, 976-979

We're grateful to the reviewer who took the time to provide their thoughts and suggestions. We have done our utmost effort to address all of the issues in the revised manuscript and believe that the updated version can meet the journal's publishing criteria.

Round 3

Reviewer 3 Report

The authors have addressed all my comments in a good manner.

I would like to thank the authors for contributing this work in our community. This paper should be published in the present form.